# Application of the Spline Interpolation in Simulating the Distribution of Phytoplankton in a Marine NPZD Type Ecosystem Model

**DOI:** 10.3390/ijerph16152664

**Published:** 2019-07-25

**Authors:** Xiaona Li, Quanxin Zheng, Xianqing Lv

**Affiliations:** 1Department of Physical Oceanography, College of Oceanic and Atmospheric Sciences, Ocean University of China, Qingdao 266100, China; 2Qingdao National Laboratory for Marine Science and Technology, Qingdao 266100, China

**Keywords:** NPZD model, phytoplankton simulation, spline interpolation, Cressman interpolation, adjoint method

## Abstract

The available observations for the model are usually sparse and uneven. The application of interpolation methods help researchers obtain an approximate form of the original data. A marine nutrient, phytoplankton, zooplankton and detritus (NPZD) type ecosystem model is applied to simulate the distribution of phytoplankton combined with the spline interpolation (SI) and the Cressman interpolation (CI). In the idealized twin experiments, the performance of these two interpolation methods is validated through the analysis of several quantitative metrics, which show the minor error and high efficiency when using the SI. Namely, the given distributions can be better inverted with the SI. The actual distribution of phytoplankton in the Bohai Sea is interpolated in the practical experiment, where a satisfactory simulation result is obtained by the model with the SI. The model experiments and results verify the feasibility and effectiveness of SI.

## 1. Introduction

With the trend of interdisciplinary and comprehensive development, studies about marine ecosystems have gradually evolved from single subjects or specific research directions (e.g., marine chemistry and biological oceanography) to multiple marine disciplines. Riley et al. [1] established a vertical one-dimensional model and quantitatively described the temporal and spatial variation characteristics of plankton in the western side of the North Atlantic Ocean, which indicated the research of marine ecology entered the era of quantitative analysis and evaluation. The development of the three-dimensional ecological dynamics model began in the 1980s [2]. For example, Kishi and Ikeda [3] set up a system of models to simulate the outbreak of red tides in the East Seto Inland; Walsh et al. [4] analyzed the fate of spring bloom within the Mid-Atlantic Bight using a three-dimensional simulation model. By the end of the 1990s, the implementation of GLOBEC (Global Ocean Ecosystem Dynamics) international programme marked the overall rise of marine ecosystem dynamical research and three-dimensional hydrodynamic-biochemical coupling models were introduced in some studies during this period [5,6,7,8].

In the aim of learning more about marine ecological characteristics, the structure of the model varies and ranges from simple types to complicated ones. The nutrient, phytoplankton and zooplankton (NPZ) model has been used for several decades and is still an important tool for plankton research, though it contains one of the simplest groups of dynamics [9]. Franks and Chen [10] coupled the NPZ model to a physical model that accurately simulated the biological patterns in the Gulf of Maine. Spatial biological parameterization was taken into a simple nutrient, phytoplankton, zooplankton and detritus (NPZD) model to improve the simulation results and sensitivity analysis was conducted to select uncorrelated and sensitive parameters [11]. Considering the pollution in Jiaozhou Bay, a three-dimensional hydrodynamic model was developed with seven components to estimate the environmental capacity of nitrogen and phosphorus [12]. Since the changing quantity and distribution of nitrogen influenced marine environment badly, Lu et al. [13] then focused on the dynamic nitrogen transference and transformation through water quality modeling in Jiaozhou Bay.

Much attention has been paid to reducing the simulation errors in many previous researches and the adjoint assimilation method is an effective method to get optimal control variables [14,15,16,17,18,19,20]. Kuroda and Kishi [21] applied the adjoint assimilation technique in a marine ecosystem model to estimate the biological parameters and choose control variables that impacted simulation results mostly. Based on Sea-Viewing Wide Field-of-View Sensor (SeaWiFS) surface chlorophyll data, Tjiputra et al. [22] optimized the ecosystem parameters in a three-dimensional global ocean biogeochemical cycle model and their experimental results indicated the applicability of the adjoint model. Yaremchuk et al. [23] proposed a hybrid program in the adjoint-free-4-dimensional data assimilation method, which searched in the subspace formed by an ensemble of directions, and its advantage was proved through experiments. The adjoint assimilation method can also be used in the marine pollutant transport model to simulate the temporal and spatial distribution of pollution [24].

Sparse and unsatisfactory observations are common in the process of scientific research. Therefore, appropriate interpolation methods are necessary to acquire accurate simulation results. Bargaoui and Chebbi [25] proposed a 3-D variogram to study the rainfall spatial variability and compared the result to a 2-D variogram, where both of the variograms were utilized as two kriging interpolation tools. In the research of Erxleben et al. [26], interpolation methods, including inverse distance weighting (IDW) and kriging, were mentioned and compared to estimate the distribution of snow in the Colorado Rocky Mountains. These two methods and the Cressman interpolation (CI) (an improved IDW method) have been adopted in various fields [11,27,28,29,30,31]. Besides, the spline interpolation (SI) performed well in obtaining smooth interpolated curves or surfaces and produced a minor error [32,33]. In our work, the SI method is applied to interpolate the distribution of phytoplankton in the Bohai Sea.

The paper is organized as follows. Section 2 describes the model, related methods and model data. The idealized twin experiments and practical experiments are carried out in Section 3. Discussion is presented in Section 4. The paper is concluded in Section 5.

## 2. Materials and Methods

### 2.1. Marine Ecosystem Model

The NPZD model used in the study considers physical, chemical and biological processes, which is a nitrogen-based model. As it shows, the model is composed of four parts: nutrient—N, phytoplankton—P, zooplankton—Z and detritus—D, where the effects of river input and atmospheric deposition on N are taken into account; the growth of P is related to temperature, nutrient and solar radiation; Z grazes P and excretes N; the mortality of P and Z increases the content of D and the remineralization of detritus forms N. The governing equations of the model are presented in Appendix A. The ecological parameters and their values are listed in Table 1 [9,10,11,34].

### 2.2. The Adjoint Model

When we evaluate a numerical model, the error between observations and simulation results is an important factor. The smaller the error is, the better the simulation result is. In addition, the adjoint assimilation method can help to reduce the error by adjusting control variables and optimizing simulation results iteratively. On the basis of cost function, the process of adjusting control variables can be conducted. The cost function, a quantitative metric to measure the mentioned error, is defined as [24,30]:(1)$$J\left(P\right)=\frac{1}{2}\int_{Τ\times \Omega }W{\left(P-P_{o}\right)}^{2}dΤd\Omega$$
where Τ and Ω are the symbols of time domain and space domain, respectively; *W* represents the weighting matrix and its elements equal to 1 when the observations are available, and 0 otherwise; *P* and *P_o_* denote the simulation results and observations of phytoplankton, respectively.

Based on the method of Lagrange multipliers, the adjoint model can be constructed. We can calculate the gradient of the cost function with respect to control variables with the adjoint model. Thus, the control variables are adjusted along the inverse direction of the gradient. In the process, the Method of Steepest Descent is adopted to modify the simulation results of phytoplankton. The iteration continues until the iteration steps run out or the requirements of error metrics (e.g., the cost function) are satisfied. The equations of the adjoint model are presented in Appendix B.

### 2.3. Independent Points Scheme and Interpolation Methods

Accurate and sufficient observations are vital for marine ecological research. However, the number of observations is not satisfactory and uneven distribution in time and space generally exists, such as discrete observations in time dimension, sparse observations in some time intervals and inevitable random and systematic errors. Therefore, the independent point scheme (IPS) is introduced in order to fit the actual situation [35,36]. Several points in the computational domain are chosen as independent points and the others are calculated by appropriate interpolation methods, which are essential for exact simulation:(2)$$p_{i,j}=\overset{N}{\underset{n=1}{\sum }}\kappa_{i,j,n}p_{n}$$
where *p_i,j_* is the value at grid (i, j); *p_n_* is the value of the *n*th independent point and N denotes the number of independent points; κ*_i,j,n_* is the weighting coefficient depending on interpolation methods. We apply the SI method in the model to simulate the distribution of phytoplankton and verify its advantages by comparing with the CI. The details of the SI and CI can be found in Guo et al. [37].

### 2.4. Data

The flow field in July 2017 of the Bohai Sea (37° N–41° N, 117.5° E–122.5° E) used in the model is provided by the Regional Ocean Modeling System (ROMS), which is a free-surface, hydrostatic, three-dimensional primitive equation regional ocean model [38]. The sea surface temperature is interpolated by the data of National Centers for Environmental Prediction (NECP). Figure 1A,B shows the sea surface flow field and temperature at 12:00 (noon), 1 July 2017. The depth of the Bohai Sea is showed in Figure 1C. The horizontal resolution is 4′ × 4′ and the model is divided into six layers vertically, whose thickness is 5 m, 10 m, 10 m, 20 m, 25 m and 25 m from top to bottom, respectively. The simulation time is 30 days with a time step of 6 h.

The initial N is the monthly mean nitrate, given by World Ocean Atlas (WOA). The distribution of P depends on whether it is used in the idealized twin experiments or the practical situation, and the specific settings are presented in Section 3. The initial values of Z and D are obtained from the model itself. Moreover, all of the state variables in the model need to be changed to nitrogen concentration with a unit mmol N m^−3^.

## 3. Results

### 3.1. Idealized Twin Experiments

Figure 2A,B shows two types of distribution of P, which are utilized in the idealized twin experiments. Considering the influence of nutrients on P and the serious pollution in Bohai Bay, Liaodong Bay and Laizhou Bay (Figure 1C), the distribution of the idealized twin experiment 1 (IE1) is prescribed as a paraboloid with high concentration in three bays and low concentration in central Bohai Sea. The distribution of the idealized twin experiment 2 (IE2) mainly takes the temperature’s influence into account.

In the idealized twin experiments, the given distributions are considered as “observations”. An initial guess of the given distributions is used as an input to solve the forward model (the NPZD model), from which the simulation results are obtained. Then, the cost function can be calculated based on Equation (1). Afterwards, run the adjoint model to compute the gradients of the cost function with respect to control variables, which are utilized to adjust the values of P. When the preset criterion is satisfied, the iterative procedure will stop and we eventually acquire the final interpolated results.

Several quantitative metrics are used to assess the performance of the SI and CI in the idealized twin experiments. The normalized cost function (NCF) is included. Besides, the mean absolute error (MAE) and the root-mean-square error (RMSE) are presented as:(3)$$\mathrm{MAE}=\frac{1}{m}\sum_{n=1}^{m}\left|X_{\mathrm{mod}}-X_{obs}\right|$$
(4)$$\mathrm{RMSE}=\sqrt{\frac{1}{m}\sum_{n=1}^{m}{\left(X_{\mathrm{mod}}-X_{obs}\right)}^{2}}$$
where *m* is the number of grids with observations; X_mod_ and X_obs_ denote the model results and given observations, respectively. The similarity coefficient (SC) can also evaluate model skill, describing the similarity degree between the model results and observations [12]:(5)$$\mathrm{SC}=1-\frac{2}{\pi }\arccos\frac{\sum_{n=1}^{m}X_{\mathrm{mod}}\times X_{\mathrm{obs}}}{\sum_{n=1}^{m}X_{\mathrm{mod}}{}^{2}\times \sum_{n=1}^{m}X_{\mathrm{obs}}{}^{2}}$$
It is better when the value of SC is closer to 1.

For IE1, the descending curves of NCF are shown in Figure 3A and the results of the quantitative metrics are listed in Table 2. The NCF using the SI declines more quickly than that using the CI and the values of NCF are reduced to 8.3 × 10^−3^ and 9.0 × 10^−2^, respectively. The result of MAE is 0.050 mmol N m^−3^ by the SI and 0.178 mmol N m^−3^ by the CI. In addition, the value of RMSE is 0.190 mmol N m^−3^ by the SI and 0.295 mmol N m^−3^ by the CI. These statistics indicate the errors are smaller when using the SI in the model. Moreover, the values of SC are 0.84 and 0.77, demonstrating the advantage of the SI over CI. Figure 4A,B depicts the simulation distributions of P with two interpolation methods, visually showing a high degree of similarity between the SI result and the given distribution (Figure 2A).

Corresponding results of quantitative metrics for IE2 are listed in Table 3. Figure 3B depicts the descending trends of NCF with two interpolation methods. The analysis for IE2 will come to the same conclusion that faster rate of decent of NCF and smaller errors produced with the SI (MAE = 0.034 < 0.120 mmol N m^−3^, RMSE = 0.096 < 0.183 mmol N m^−3^) show its advantage over the CI. Besides, a same characteristic as IE1 is obtained by comparing the similarity degree (SI: SC = 0.88, CI: SC = 0.78). The distributions of P inverted with the adjoint model combined with two interpolation methods are presented in Figure 5A,B, also illustrating the simulation result with the SI is consistent with the given distribution (Figure 2B).

### 3.2. Practical Experiment

In this section, the P data is converted from the monthly mean surface chlorophyll-a (chl-a) provided by SeaWiFS, which is a satellite-borne sensor to collect global ocean biological data [39,40]. The complete distribution of P in the Bohai Sea is interpolated with interpolation methods. Each state variable needs to be converted to nitrogen concentration, but for chl-a (mg m^−3^), it needs to be converted to carbon (mg m^−3^) firstly with the following equation [41,42]:(6)$$C=\rho_{\max}\frac{\mathrm{chl}-a}{\mathrm{chl}-a+K_{1/2}}\mathrm{chl}-a$$
where ρ_max_ = 90 and the half-saturation coefficient *K*_1/2_ = 0.477. Then, the Redfield ratio of C/N (106 mol C/16 mol N) is used to change carbon to nitrogen.

In the idealized twin experiments, it has been proved that the SI method performs well comparing to the CI in simulating the distribution of P. Therefore, we can implement the SI method to interpolate practical observed data and also compare the interpolated results with CI. The model results using the SI and CI are shown in Figure 6A,B. The concentration of P in the three bays is higher than that in the centre of the Bohai Sea. The distribution pattern is exactly in accordance with the pollution in the Bohai Sea, where the pollution in the three bays is more serious than that in the center [30]. Figure 7 shows the descending curves of NCF, indicating the performance of the model combined with the SI in simulating P is superior to that with the CI.

## 4. Discussion

To acquire satisfactory simulation results, it is essential to choose an appropriate interpolation method. Actually, previous studies have compared the characteristics of several interpolation methods [26,29,32]. Pan et al. [32] proposed a curve interpolation experiment to compare the kriging interpolation, the SI and the CI, showing the accuracy and smoothness of the SI result. In the experiments, we have testified the accuracy and efficiency with the SI method. In order to explain its smoothness clearly, an experiment is conducted here. The prescribed surface is presented in Figure 8A. Then, we apply the SI and CI to obtain the interpolation results. Based on the comparison of the SI and CI results (Figure 8B,C), the smoothness of the former one is demonstrated obviously. Therefore, the simulation result obtained by the SI has better accuracy, efficiency and smoothness.

## 5. Conclusions

Sparse and unsatisfactory observations have been a constraint for multifaceted exploration of marine ecology. To fit the status, IPS has been applied to compute an approximate distribution. In this way, the application of interpolation methods is an important factor in reducing simulation errors. Among them, CI is a useful tool and has been widely adopted in previous work. However, the weakness of the CI that affects the accuracy of simulation result exists. Improving the accuracy is exactly one of the advantages of the SI. In the study, the performance of the SI method is testified in simulating the distribution of P.

In the idealized twin experiments, the metric NCF shows the high efficiency of the SI compared with the CI. In addition, a minor error can be achieved for the SI through the analysis of MAE and RMSE. The metric SC demonstrates the model result obtained by using the SI is closer to the given distribution. In the practical experiment, the model combined with two interpolation methods is applied to interpolate the actual distribution of P based on the incomplete observations. The quickly descending curve of NCF illustrates the interpolated distribution by the SI is convincing and satisfactory.

According to the experiments we implement, the application of the SI method can also improve the smoothness of the interpolation result. Consequently, it is worth trying to use the SI method in many other aspects, such as the simulations of other variables and researches about more complicated ecosystem models.

## Figures and Tables

**Figure 1 ijerph-16-02664-f001:**
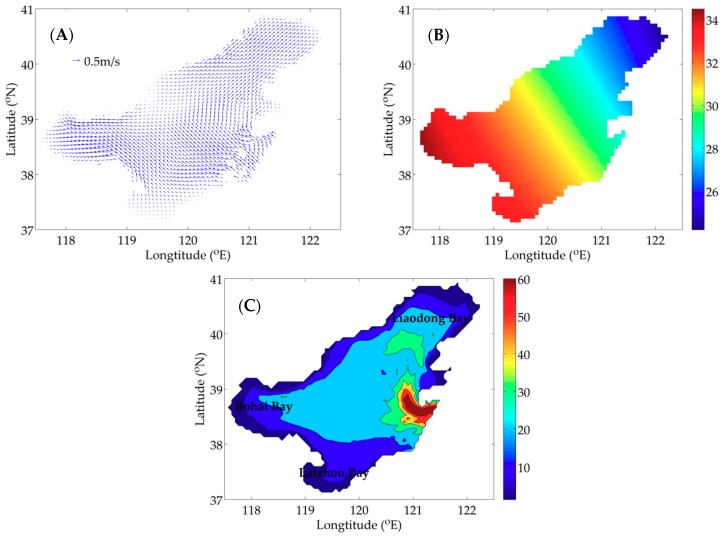
The surface flow field (**A**), the sea surface temperature (**B**) (unit: °C) at 12:00 (noon), 1 July 2017 and the depth (**C**) (unit: m) of the Bohai Sea.

**Figure 2 ijerph-16-02664-f002:**
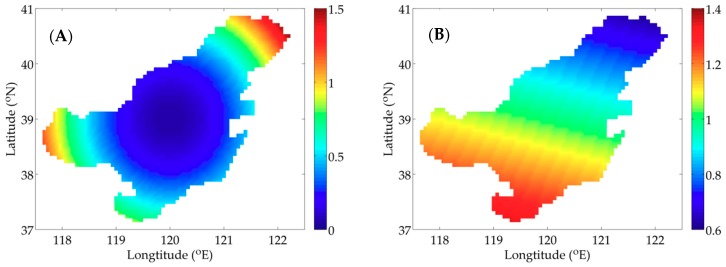
Given distributions of P in the idealized twin experiment 1 (IE1) (**A**); and the idealized twin experiment 2 (IE2) (**B**) (unit: mmol N m^−3^).

**Figure 3 ijerph-16-02664-f003:**
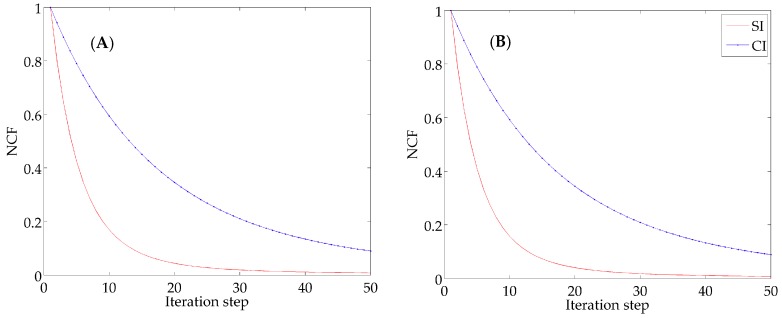
The descending curves of the normalized cost function (NCF) using two interpolation methods in IE1 (**A**) and IE2 (**B**).

**Figure 4 ijerph-16-02664-f004:**
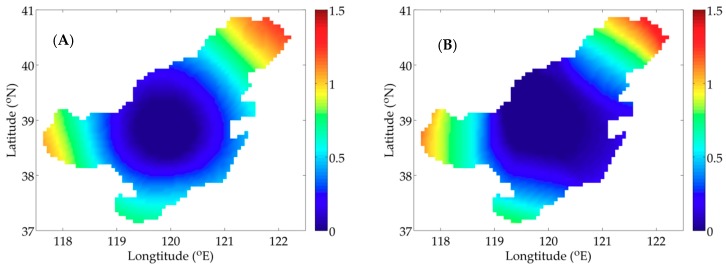
Distributions of P inverted with the adjoint method combined with the spline interpolation (SI) (**A**) and the Cressman interpolation (CI) (**B**) in IE1 (unit: mmol N m^−3^).

**Figure 5 ijerph-16-02664-f005:**
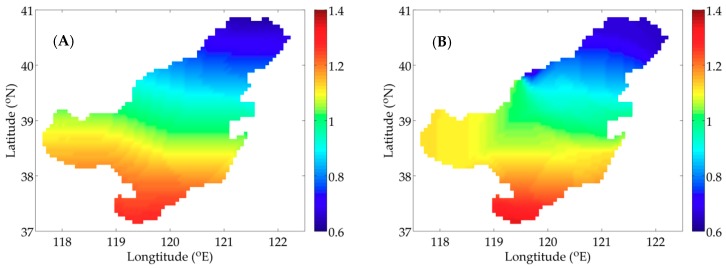
Distributions of P inverted with the adjoint method combined with SI (**A**); and CI (**B**) in IE2 (unit: mmol N m^−3^).

**Figure 6 ijerph-16-02664-f006:**
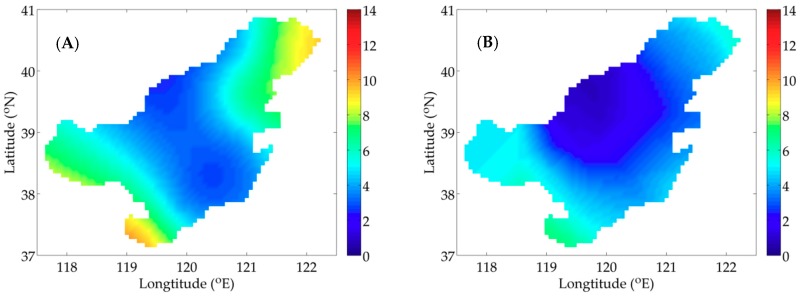
Distributions of P inverted with the adjoint method combined with SI (**A**); and CI (**B**) in practical experiment (unit: mmol N m^−3^).

**Figure 7 ijerph-16-02664-f007:**
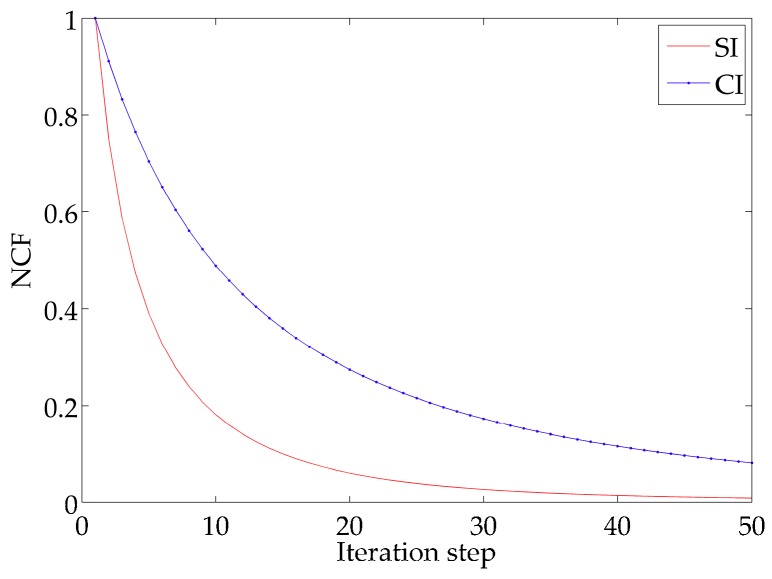
The descending curves of NCF using two interpolation methods in practical experiment.

**Figure 8 ijerph-16-02664-f008:**
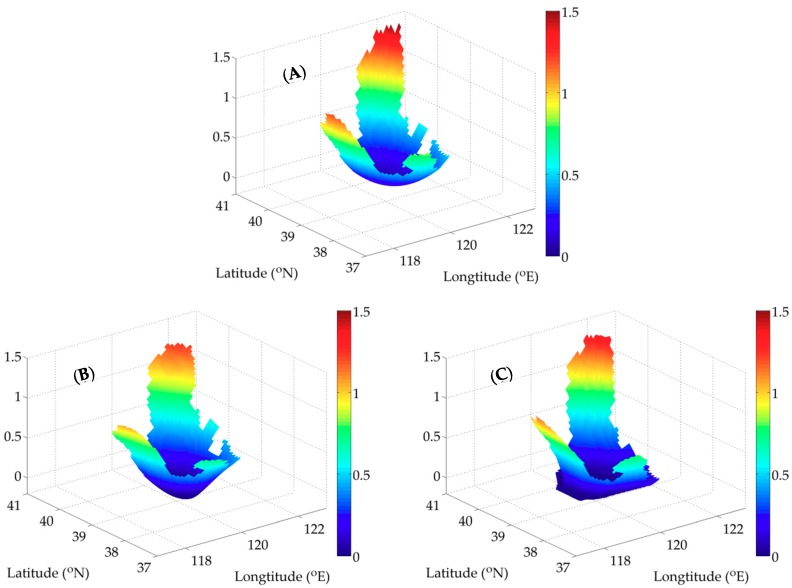
Comparison of the interpolation results by the SI and CI. (**A**) The prescribed surface; (**B**) the interpolation result by the SI; (**C**) the interpolation result by the CI (unit: mmol N m^-3^).

**Table 1 ijerph-16-02664-t001:** Ecological parameters and their values in the model.

Parameter	Symbol	Value	Unit
Maximum growth rate of phytoplankton	*V_m_*	1.0	day^−1^
Maximum grazing rate of zooplankton	*G_m_*	0.5	day^−1^
Mortality rate of phytoplankton	*D_p_*	0.1	day^−1^
Mortality rate of zooplankton	*D_z_*	0.2	day^−1^
Remineralization rate of detritus	*e*	0.05	day^−1^
Temperature coefficient for phytoplankton growth at 10 °C	*AQ* _10_	2.08	-
Temperature coefficient for zooplankton growth at 10 °C	*BQ* _10_	3.10	-
Assimilation ratio of zooplankton	*γ*	0.75	-
Excretion ratio of zooplankton	*θ*	0.03	-
Attenuation coefficient of light	*K_ext_*	1.0	m^−1^
Optimum irradiance	*I_o_*	100	W m^−2^
Sinking velocity of phytoplankton	*w_p_*	0.73	m day^−1^
Sinking velocity of detritus	*w_d_*	1.00	m day^−1^
Half-saturation constant for nutrient uptake	*K_s_*	1.0	mmol m^−3^
Ivlev constant of zooplankton	*f*	0.2	m^3^(mmol N)^−1^

**Table 2 ijerph-16-02664-t002:** The results of the quantitative metrics using two interpolation methods in the idealized twin experiment 1 (IE1).

Interpolation Method	NCF	MAE (mmol N m^−3^)	RMSE (mmol N m^−3^)	SC
SI	8.3 × 10^−3^	0.050	0.190	0.84
CI	9.0 × 10^−2^	0.178	0.295	0.77

**Table 3 ijerph-16-02664-t003:** The results of the quantitative metrics using two interpolation methods in the idealized twin experiment 2 (IE2).

Interpolation Method	NCF	MAE (mmol N m^-3^)	RMSE (mmol N m^-3^)	SC
SI	7.8 × 10^-3^	0.034	0.096	0.88
CI	8.9 × 10^-2^	0.120	0.183	0.78

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
