# Peer review of "Application of the Spline Interpolation in Simulating the Distribution of Phytoplankton in a Marine NPZD Type Ecosystem Model"

_ijerph, 2019, doi:10.3390/ijerph16152664_

Round 1
Reviewer 1 Report
Comments to authors
General comments
Phytoplankton are important organisms in oceanic waters because of their role in marine ecosystem and biogeochemical cycling. In this study, the authors investigated the differences between two interpolation methods (CI and SI), what is very useful in environmental studies considering the gaps that can exist in field data. However, although there is a detailed mathematical description of the model and interpolation methods used, the manuscript lack of coherence in many parts regarding the interpretation of results. I feel that probably English was a problem and I recommend the authors to consider spending a little bit more time working on the English and making sure that the sentences are clear and the message of this paper is straightforward. In many parts of the documents (marked in yellow along the manuscript) it is difficult to understand what the authors want to say. Below I indicate some specific comments and I indicated along the pdf some points where I consider the English was very confusing and made difficult to be sure about discussion. Thus, I recommend major modifications, given that the manuscript is not suitable for publication in the current state.
Specific comments
Line 86: I suggest to remove the expression "As everybody knows".
Line 100-101: use "the number of observations" instead of "the quantity".
Line 109-110: Why do you mean with applying CI as a contrast? Please, add some explanation.
Line 116: Figure 1a,b shows ...
Figure 1. Use larger letters and fonts inside the figures, i.e. (a), (b) and (c).
Line 129. Two types of given distribution of P, ... Consider reviewing the English ... Probably the entire sentence should be rewritten. It is confusing.
Lines 136 to 141. English is also confusing. For example, "Then give an initial guess of the given distribution to solve .... and obtain the simulation results". It is more appropriate to say: Then a initial guess of the given distributions were used as input to solve the forward model ..." The entire paragraph should be rewritten.
Line 149. Please, add a reference.
Line 159. the term: "high closeness degree" is confusing. Do you mean that results are similar? What does it mean?
Figure 4 and 5 legend. Please add letters to figures and indicate that in the legend.
Line 175: ...P data WAS OBTAINED from chlorophyll-a provided by SeaWiFS. Please, insert in methodology information about SeaWiFS data (processing level, resolution, algorithm used, etc ...).
Paragraph L181-188: The first phrase (In idealized twin experiments, ...) is not clear. Are you talking about your data or from other investigations? Then, when you say "in a practical context" you mean the use of Chla data from SeaWiFS? It is not clear what you are talking about here. Finally, SI is acceptable because it is better than CI? Ok I agree that it is better, but what you mean with acceptable? Acceptable for what? You mention that results are in accordance with the "actual situation" but in figure 7 SI results indicate higher values of N to the north while CI did not. It is also higher to the south of the bay. You have to explain this.
Line 196-197: Please consider rewrite "However, large errors exist and the model results using the CI is not smooth enough" ... you have been already talking about CI in the previous phrase. In general, conclusion lacks of a clear message and some statements repeat what was said in discussion without more insights in the results obtained.
Line 200: "In THE idealized twin experiments ... : the lack of the word "THE" gives the idea that you are talking about any twin experiments and not exactly about yours. This is confusing in many parts of the text.
Reviewer 2 Report
The authors reported the application of the spline interpolation in simulating phytoplankton in a marine ecosystem model. The topic can be interesting and, to some degree, to be meaningful. However, the reviewer has several concerns:
The present title is too simple and abstract, and it lacks the specific aims and details. Is the application of spline interpolation in marine ecosystem model new? and the title also needs to specify what aspects of marine phytoplankton they are working one. Third, which marine ecosystem model? Please specify these details.
The language can be further improved. Some subjective viewpoints and essay-like words should not be used in the scientific work. For example, " As everybody knows, the smaller the errors between the observations and modeled values are," the sentence of As everybody knows should be edited. ....
Table 1 lists the ecological parameters and their values, is this model totally new or is it built previously? if the latter, please add the citation. Meanwhile, please briefly explain these values of some parameters in the model. For example, the max. of phytoplankton growth rate is 1 day(-1), doese this value make scientific sense?
The manuscript lacks the sufficient discussion. At least it should discuss the strength and weakness of the modified model.
Round 2
Reviewer 1 Report
Manuscript improved. Thank you for considering my recommendations.
Reviewer 2 Report
I have no further big concerns about the current version.